ⓐ | **Open Peer Review** | Antimicrobial Chemotherapy | Research Article

# Whole-genome analysis of extensively drug-resistant *Acinetobacter baumannii* isolates from a Peruvian tertiary hospital reveals the emergence of OXA-23-producing ST79 and ST1079 clones

Jesus G. M. Pariona,[1] Heli Barrón-Pastor,[1] José E. Tinedo del Aguila,[1] David Santos-Lázaro,[1] Doris Huerta-Canales,[1] Mario Monteghirfo-Gomero,[1] Carolina Cucho-Espinoza,[1,2] Luz Huaroto-Valdivia,[1,2] Yesica Llimpe Mitma de Barrón[1]

**ABSTRACT** *Acinetobacter baumannii* is a leading cause of extensively drug-resistant (XDR) infections in intensive care units (ICUs), with global concern due to its resistance to nearly all antimicrobials. In 2024, the WHO reaffirmed carbapenem-resistant *A. baumannii* as a critical-priority pathogen. In Peru, over 97% of clinical *A. baumannii* isolates are carbapenem non-susceptible, yet the genomic features of local strains remain poorly characterized. We performed whole-genome sequencing of 19 XDR *A. baumannii* isolates collected from ICU patients at Hospital Nacional Dos de Mayo (Lima, Peru) between May 2023 and September 2024. Genomes were analyzed for sequence types (ST), capsular (KL) and lipooligosaccharide (OCL) loci, virulence and resistance genes, and phylogenetic relatedness. All isolates were XDR but remained susceptible to colistin. Fourteen isolates belonged to ST2, forming two closely related subgroups carrying KL2 or KL9 and OCL1, with low SNP distances, suggesting recent clonal expansion. These ST2 genomes lacked $bla_{OXA-23}$ but carried $bla_{OXA-72}$. Two isolates were assigned to ST79, harboring KL13/OCL10 and $bla_{OXA-23}$, representing the first report of this resistance gene in Peruvian ST79 strains. Three isolates belonged to ST1079, forming a tight Peruvian clade genetically distant from public ST1079 genomes; all carried KL107/OCL3 and $bla_{OXA-23}$, indicating possible local emergence. Our study revealed the emergence in Peru of XDR *A. baumannii* lineages ST79 and ST1079 harboring $bla_{OXA-23}$ and the spread of high-risk ST2 clones carrying $bla_{OXA-72}$, underscoring the need for enhanced genomic surveillance and targeted infection-control measures.

**IMPORTANCE** *A. baumannii* is one of the most problematic hospital pathogens worldwide, often resistant to nearly all available antibiotics. In Peru, the proportion of carbapenem-resistant isolates is among the highest reported globally, yet their genetic background has remained largely unknown. This study provides the first genomic and phylogenomic insight into XDR *A. baumannii* from a major Peruvian hospital, revealing the spread of high-risk clones belonging to sequence type 2 carrying the OXA-72 carbapenemase, and the emergence of two additional lineages, sequence type 79 and sequence type 1079, both producing OXA-23. These findings demonstrate that multiple resistant lineages are established in Peru, highlighting the urgent need to implement genomic surveillance and infection control measures. Understanding the diversity and dynamics of these lineages is critical to limit their further spread and to guide public health responses in South America.

**KEYWORDS** antimicrobial resistance, carbapenemase, genomic surveillance, clonal dissemination, nosocomial infections, Latin America

Address correspondence to Jesus G. M. Pariona, mjesusgiovani@gmail.com, or Yesica Llimpe Mitma de Barrón, yllimpem@unmsm.edu.pe.

The authors declare no conflict of interest.

See the funding table on p. 11.

Extensively drug-resistant (XDR) *Acinetobacter baumannii* has become a major public health threat in healthcare settings worldwide. Characterized by resistance to nearly all clinically available antibiotics, XDR *A. baumannii* poses serious therapeutic challenges, especially in intensive care units (ICUs), where it is frequently associated with severe infections and high mortality rates (1–4). Its remarkable ability to survive under harsh environmental conditions, tolerate desiccation, and acquire resistance genes contributes to its persistence in hospitals and its role in nosocomial outbreaks (1, 2).

For carbapenem-resistant strains, often a hallmark of the XDR phenotype, mortality can range from 16% to 76%, with outcomes significantly worse than those associated with carbapenem-susceptible isolates (4). Resistance to carbapenems has been documented in 19–67% of isolates in clinical studies, underscoring the limited treatment options available (4). As a result, the World Health Organization (WHO) has listed carbapenem-resistant *A. baumannii* as a critical-priority pathogen, a designation reaffirmed in 2024 to emphasize the need for new antimicrobial agents and stronger global surveillance (5).

Globally, carbapenem resistance in *A. baumannii* is largely mediated by the horizontal acquisition of class D β-lactamase genes, particularly $bla_{OXA-23}$, which accounts for resistance in over 80% of carbapenem-resistant *A. baumannii* isolates (6). Other resistance determinants include $bla_{OXA-24/40}$, $bla_{OXA-58}$, and metallo-β-lactamases like $bla_{NDM-1}$ (6).

In Latin America, the epidemiological burden of XDR *A. baumannii* is among the highest globally. Multicenter studies have reported carbapenem non-susceptibility rates ranging from 50% to over 85% in countries such as Argentina, Brazil, and Chile (7, 8). Surveillance efforts between 2013 and 2014 identified $bla_{OXA-23}$ in nearly all carbapenem-resistant isolates across six South American countries, with a predominance of international clones (ICs) such as IC5 (Pasteur ST79) and IC4 (ST15), and more recent expansion of IC2 (ST2) across the region (7–13). In Brazil, 73.2% of carbapenem-resistant isolates carried $bla_{OXA-23}$, while in Argentina, this gene was found in 100% of isolates from 2016 to 2017 (8, 9).

Despite this regional crisis, genomic data from Peru remain strikingly scarce. Laboratory-based surveillance conducted through the Latin American Antimicrobial Resistance Surveillance Network (ReLAVRA) reported that, in 2013, carbapenem resistance in *Acinetobacter* spp. in Peru reached 78%, one of the highest rates in Latin America (14). Additional hospital-based studies conducted in Lima between 2014 and 2016 confirmed alarmingly high resistance rates, with up to 97% of *A. baumannii* isolates exhibiting carbapenem resistance (13). Nevertheless, only eight Peruvian *A. baumannii* genomes are publicly available, limiting our understanding of the dominant resistance mechanisms, circulating clonal lineages, and their phylogenetic relationships (https://www.ncbi.nlm.nih.gov/pathogens/isolates/#taxid:470; accessed on May 11, 2025).

To address this, we performed whole-genome sequencing (WGS) of 19 phenotypically confirmed XDR *A. baumannii* isolates collected from patients at Hospital Nacional Dos de Mayo, a large tertiary care center in Lima, Peru. Our study aimed to characterize the genetic background, resistance genes, and clonal structure of these isolates, thereby contributing critical data to the underrepresented genomic landscape of *A. baumannii* in the country and region.

## MATERIALS AND METHODS

### Sample processing and species identification

A total of 19 clinical isolates were obtained from the microbiology laboratory of Hospital Nacional Dos de Mayo, a healthcare facility with more than 700 beds. The collection period was from May 2023 to September 2024. During this period, all isolates identified as *A. baumannii* exhibiting an XDR profile were included in the study. The selection was performed by convenience sampling, including all non-redundant isolates recovered from clinical samples, regardless of the type of specimen, as long as they met the

XDR profile. Bacterial isolation was performed in MacConkey and blood agar. Identification of *A. baumannii/calcoaceticus* complex was performed through matrix-assisted laser desorption/ionization-time of flight mass spectrometry (MALDI-TOF/MS), using the Bruker MALDI Biotyper system (software version 5.0.2) and the BD Phoenix Panel. An identification at the species level was considered a score ≥2.0. Based on MALDI-TOF identification, a single, isolated colony of presumptive *A. baumannii* was selected and streaked onto a Mueller–Hinton (MH) agar plate (Oxoid, Hampshire, UK), then incubated again at 37°C for 24 h. A single, isolated colony from the MH plate was selected and stored for downstream analyses.

## Antimicrobial susceptibility profiling

Minimum inhibitory concentrations (MICs) of ampicillin-sulbactam, piperacillin-tazobactam, ceftazidime, cefepime, imipenem, meropenem, gentamicin, amikacin, ciprofloxacin, levofloxacin, trimethoprim-sulfamethoxazole, and colistin were evaluated by the BD Phoenix AST Panel, according to the Clinical and Laboratory Standards Institute 2021 guidelines and European Committee on Antimicrobial Susceptibility Testing (EUCAST) 2024 breakpoint tables (15, 16). The XDR was defined as nonsusceptibility to at least one agent in all but two or fewer antimicrobial categories (i.e., bacterial isolates remain susceptible to only one or two antimicrobial categories) (17).

## Illumina sequencing

All collected isolates were subjected to Illumina WGS. Briefly, DNA was extracted using the Nucleic Acid Extraction Kit, magnetic bead method, using the EXM 3000 Nucleic Acid isolation System (ZYBIO). Libraries were constructed using the DNA Prep Kit (Illumina, San Diego, CA) according to the manufacturer's protocol. Library concentrations were quantified using a Promega Quantus Fluorometer (Promega Corporation). Paired-end sequencing (2×150 bp reads) was performed on the Illumina iSeq100 platform (Illumina Inc., San Diego, CA, USA).

## Bioinformatic analyses

The quality of raw sequencing data were assessed using FastQC v0.12.1 (https://github.com/s-andrews/FastQC). Short-read sequences were trimmed using Trimmomatic v0.39 (https://github.com/usadellab/Trimmomatic), and *de novo* assembly was performed with Unicycler v0.4.8 (https://github.com/rrwick/Unicycler). Assembly quality was assessed with QUAST v5.0.2 (https://github.com/ablab/quast), and species identity was confirmed using FastANI v1.33 (https://github.com/ParBLiSS/FastANI). Potential contamination was evaluated using Kraken2 v2.0.7 (https://github.com/DerrickWood/kraken2; using standard database).

The resistome was predicted using ResFinder v4.4.2 (https://github.com/tseemann/abricate). Multi-locus sequence typing (MLST) was performed according to the Pasteur scheme using MLST v2.23.0 (https://github.com/tseemann/mlst). Virulence genes were detected using ABRicate v1.0.1 (https://github.com/tseemann/abricate) with the Virulence Factors Database (VFDB) database, and efflux pump genes were predicted using the Comprehensive Antibiotic Resistance Database (CARD, https://card.mcmaster.ca/). Mutations in genes related to fluoroquinolone resistance were predicted by CARD. Capsular polysaccharide (KL) and the outer core of lipooligosaccharide (OCL) were predicted using Kaptive (https://github.com/klebgenomics/Kaptive). All predictions were performed using a ≥95% identity threshold.

For phylogenetic analysis, core-genome alignments were predicted with Roary v.3.13.0 (https://github.com/sanger-pathogens/Roary). Polymorphic sites were extracted with Gubbins v2.4.0 (https://github.com/nickjcroucher/gubbins), excluding those that were predicted to occur via recombination. Core-genome SNPs (cgSNPs) were extracted from the core gene alignment using SNP-sites v2.5.1 (https://github.com/sanger-pathogens/snp-sites).

In addition, all publicly available *A. baumannii* genomes from Peru deposited in the National Center for Biotechnology Information (NCBI) database (retrieved on May 11, 2025) were included in the phylogenetic reconstruction.

A maximum-likelihood phylogenetic tree was inferred using RAxML-NG v1.2.0 (https://github.com/amkozlov/raxml-ng) under the GTR+Gamma substitution model with 1,000 bootstrap replicates to assess branch support. Pairwise SNP distances were calculated with snp-dists v0.8.2 (https://github.com/tseemann/snp-dists), and the resulting phylogenetic tree was visualized and annotated using iTOL v6 (https://itol.embl.de/).

## Global phylogeny for ST2, ST79, and ST1079

To construct global phylogenies for *A. baumannii* clones ST2, ST79, and ST1079 isolated in this study, we downloaded 45,112 *A. baumannii* genome assemblies from the NCBI database (search conducted on May 11, 2025). All *A. baumannii* genomes were analyzed using MLST v2.23.0 to determine sequence types (STs) and ABRicate v1.0.1 with the Resfinder 4.0 database to detect carbapenem resistance genes.

For the ST2 phylogeny, 200 genomes were selected, including one representative genome sequenced in this study and 199 global genomes showing the closest SNP distances to full-length whole-genome H2M2302 (range: 36–129 SNPs, Table S1). This selection aimed to include the most phylogenetically related isolates to provide a high-resolution context for the Peruvian strain while avoiding redundancy and ensuring computational feasibility. Importantly, this cutoff was chosen to provide lineage-level context, not to infer recent transmission events, which are typically characterized by much lower SNP distances (≤2–3 core SNPs) among outbreak-related isolates (18). Genome similarity between public genomes and one representative genome from this study (H2M2302) was estimated using Snippy v4.6.0. Thus, genome-wide SNP distances were used to select the closest related isolates. These assemblies were then subjected to a core-genome alignment workflow: gene clustering was performed with Roary v3.13.0, cgSNPs were extracted using SNP-sites v2.5.1 (1–77 SNPs relative to H2M2302, Table S2), and maximum-likelihood phylogenetic inference was carried out with RAxML-NG using the same parameters as described in the previous section.

For the global phylogenies of ST79 and ST1079, all publicly available genome assemblies in the NCBI database were used. Specifically, 280 genomes (two study strains and 278 publicly available genomes) were included for ST79, and 6 genomes (three study strains and three from NCBI) were included for ST1079. In both cases, core-genome alignments were generated as above, and phylogenies were inferred using the RAxML-NG parameters as described in the previous section. Final phylogenetic trees were visualized using iTOL v6.

## RESULTS

### All ICU *A. baumannii* isolates were extensively drug-resistant

The 19 XDR *A. baumannii* isolates were obtained from 18 patients hospitalized between May 2023 and September 2024. The patient cohort comprised 15 males and 3 females, with ages ranging from 10 to 80 years. Most patients (14/18) were admitted to intensive care units, including general, cardiovascular, neurological, and pediatric ICUs. The isolates were primarily recovered from lower respiratory tract specimens (13/19, 68.4%), including bronchial secretions ($n = 7$), bronchoalveolar lavage ($n = 4$), and sputum ($n = 1$). Other sources of infection included blood ($n = 2$), cerebrospinal fluid ($n = 2$), urine ($n = 1$), an abscess ($n = 1$), and necrotic tissue from a wound ($n = 1$).

Antimicrobial susceptibility testing revealed a uniformly high level of resistance among all *A. baumannii* isolates. All strains were resistant to ceftazidime, imipenem, meropenem, piperacillin/tazobactam, trimethoprim/sulfamethoxazole, fluoroquinolones, and gentamicin (100%) (Fig. 1A). Resistance to amikacin was also frequent, observed in 18 of 19 isolates (94.7%). For cefepime, 12 isolates (63.2%) were resistant,

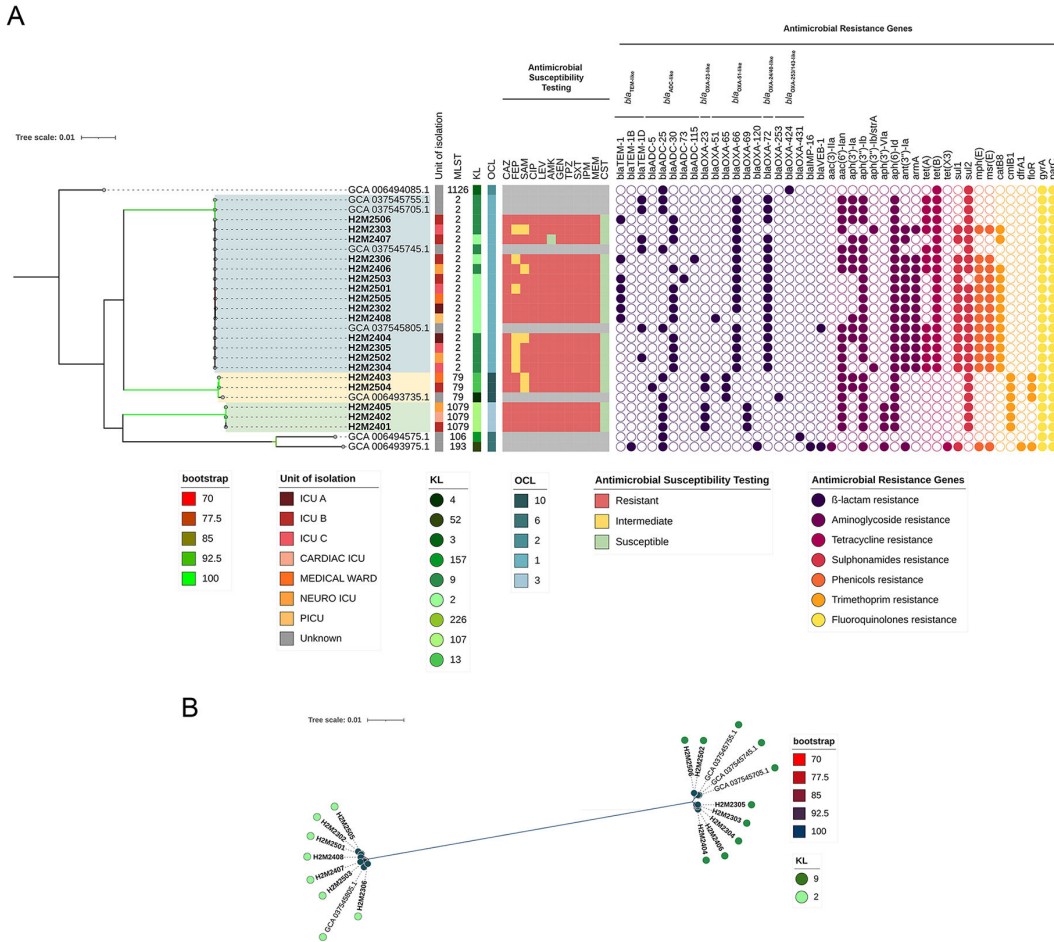

**FIG 1** (A) Core-genome phylogeny of *A. baumannii* isolates and publicly available genomes from Peru, constructed using single-nucleotide polymorphisms identified from whole-genome alignments. The accompanying metadata display the antimicrobial susceptibility profiles, sequence types (STs), KL and OCL locus types, and the presence of acquired antimicrobial resistance genes (resistome). Bootstrap support values ranging from 70% to 100% are indicated on the branches. (B) Core-genome-based phylogeny of *A. baumannii* isolates from this study belonging to sequence type 2 (ST2), showing the distribution of KL loci and their genetic relationships among isolates recovered from the Peruvian ICU. Bootstrap support values (70–100%) are displayed on the branches.

and 7 (36.8%) exhibited intermediate susceptibility. Similarly, 14 isolates (73.7%) were resistant to ampicillin/sulbactam, while 5 (26.3%) showed intermediate susceptibility. Notably, all isolates remained susceptible to colistin. Based on these profiles, all isolates were classified as XDR, according to standardized antimicrobial susceptibility criteria.

## Phylogenomic structure of *A. baumannii* isolates from Peru

Genomic data from all sequenced *A. baumannii* isolates (designated "H2M") were supplemented with eight additional publicly available Peruvian genomes from NCBI to provide a broader context (Fig. 1A). All assemblies passed quality control and were confirmed as *A. baumannii*. Peruvian genomes belonged to ST2 (14 study isolates + 4 public genomes), ST79 (2 study isolates + 1 public genome), ST1079 (3 study isolates), ST106 (1 public), ST193 (1 public), and ST1126 (1 public).

KL and OCL locus analyses revealed that ST2 isolates predominantly carried KL2 and KL9 with OCL1, ST79 harbored KL13/OCL10, and ST1079 possessed KL107/OCL3, reflecting lineage‑specific surface polysaccharide repertoires (Fig. 1A).

Resistome profiling identified *bla*<sub>OXA-23</sub> exclusively within the ST1079 clade (3/3 isolates) and the ST79 clade (2/3 isolates), whereas ST2 genomes carried the intrinsic

*bla*~OXA-51-like~ and *bla*~TEM-1~ alleles and acquired *bla*~OXA-72~, which is associated with a variant of the OXA-24/40 family known to mediate carbapenem resistance (Fig. 1A).

Additional resistance determinants included the 16S rRNA methyltransferase *armA* and aminoglycoside-modifying enzymes (*aac*(6′)-*Ib3*, *aph*(3′′)-*Ib*, *aph*(3′)-*Ia*, *aph*(3′)-*VI*, *aph*(6)-*Id*), along with tetracycline efflux pumps (*tet*(A), *tet*(B)), the sulfonamide resistance gene (*sul1*), and point mutations in the fluoroquinolone target genes *gyrA* and *parC*. On the other hand, virulence genes associated with eight functional categories, including adherence, biofilm formation, quorum sensing, and secretion systems, exotoxins, immune evasion, iron uptake systems, and two-component systems were detected in all isolates (Table S3).

The cgSNP-based phylogeny resolved three well-supported clades corresponding to ST2, ST79, and ST1079 (Fig. 1A; Table S4). In the pairwise core-genome phylogenetic analysis of ST2 *A. baumannii*, two monophyletic subgroups associated with different *KL* locus types were identified (Fig. 1B). Within the ST2/KL2 subgroup, genome-wide SNP distances ranged from 7 to 57 SNPs (mean ± SD: 33.00 ± 11.80 SNPs), while the ST2/KL9 subgroup showed pairwise SNP distances from 3 to 49 SNPs (mean ± SD: 30.93 ± 13.56 SNPs), both indicative of low genetic diversity and consistent with recent clonal expansion (Table S5). Notably, three public ST2 genomes from Ayacucho (Andean region) clustered within the KL9 subgroup, while a fourth genome from Amazonas (Amazon region) grouped with the KL2 subgroup. In contrast, the ST79 clade displayed wider genetic variability, with SNP distances ranging from 46 to 2267 (mean ± SD: 1855.67 ± 1017.98 SNPs), reflecting the presence of a single local cluster (46 SNPs) and a genetically distant public genome. On the other hand, ST1079 lineages exhibited low pairwise SNP distances (7–27 SNPs; mean ± SD: 20.33 ± 9.24 SNPs), supporting the recent local emergence of this clone.

## Global phylogenomic context of ST2 *A. baumannii*

The global phylogeny resolved multiple deeply branching clades within ST2, placing H2M2302 within a distinct South American sublineage that lacked the OXA-23 carbapenemase determinant (Fig. 2). All genomes were uniformly typed as capsule locus KL2 and outer-core locus OCL1 (Fig. 2).

Public metadata analysis showed that 97.8% of genomes were from the Americas (85% USA, 8% Mexico, 4% Ecuador, 1% Peru, and 0.5% Canada), 1.6% from Asia (China), and 0.5% from Europe (France), reflecting major widespread continental dissemination (https://www.ncbi.nlm.nih.gov/pathogens/isolates/#taxid:470). Moreover, 65.5% of genomes within this phylogeny also lacked OXA-23, highlighting the rise of a non-OXA-23–producing ST2 subclone.

## Global phylogenomic context of ST79 and ST1079 *A. baumannii*

The global phylogenetic analysis of ST79 revealed deep-branching clades that corresponded closely to geographical provenance and genetic markers (Fig. 3; Table S6). Among the 280 genomes analyzed, 94.6% carried the outer-core locus OCL10, with the remainder distributed among OCL1, OCL3, OCL5, OCL6, and OCL7. Capsule locus diversity was high (25 KL subtypes), yet KL151 and KL9 predominated, representing 29.3% and 26.4% of genomes, respectively (Fig. 3).

Geographical public metadata revealed a pronounced American bias: 39.2% of genomes originated from the USA and 34.2% from Brazil, followed by Paraguay (7.9%), Argentina and Chile (each 1.8%), Colombia (1.4%), South Africa and Romania (each 1.4% and 2.9%, respectively), and small fractions from Bolivia, Ecuador, Ukraine, and other countries (each ≤1.1%) (https://www.ncbi.nlm.nih.gov/pathogens/isolates/#taxid:470). Peruvian study strains clustered tightly with contemporaneous isolates from the USA and other South American countries, supporting a regional sublineage within the broader American radiation.

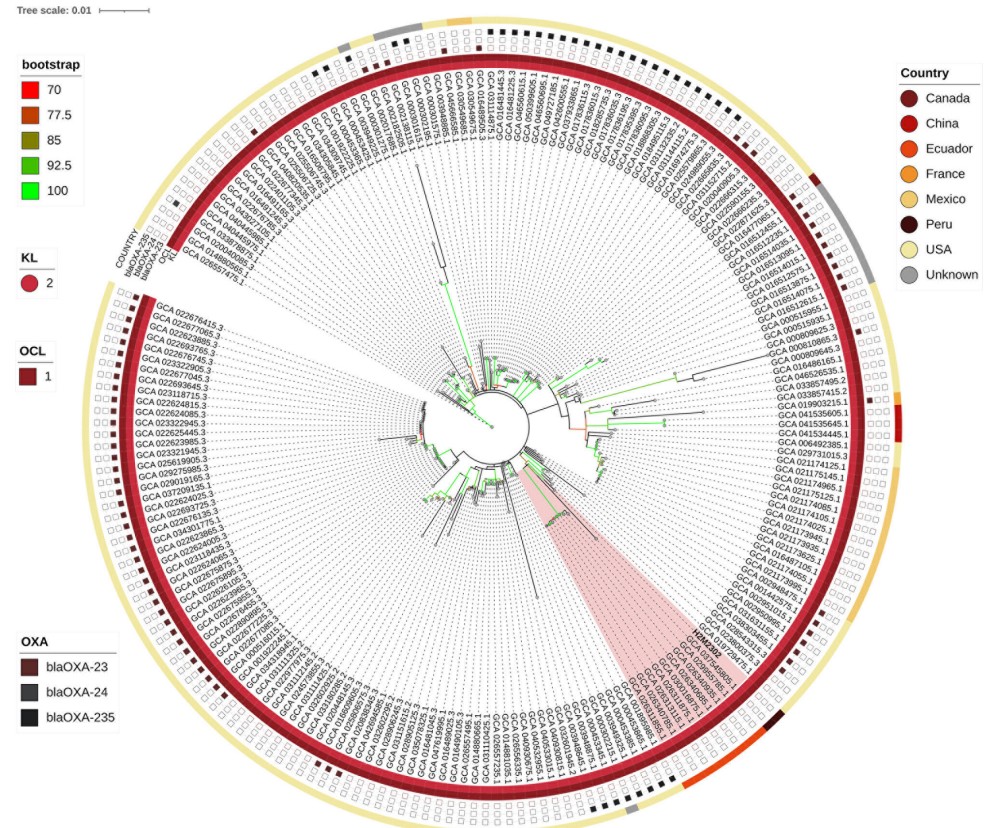

**FIG 2** Global phylogenetic relationships of *A. baumannii* ST2 isolates. A maximum-likelihood tree constructed from cgSNPs illustrating the global distribution and relationships of *A. baumannii* ST2 isolates. Branch support values (bootstrap, 70–100%) are indicated along the branches. Metadata tracks indicate KL and OCL locus types, country origin, as well as the presence of major acquired carbapenemase genes ($bla_{OXA-23}$, $bla_{OXA-24}$, $bla_{OXA-235}$, and $bla_{NDM-1}$). The Peruvian isolates sequenced in this study are highlighted to show their phylogenetic placement within the global ST2 population.

Carbapenemase profiles further stratified the tree: 43.2% of ST79 genomes encoded OXA-23, primarily in Brazilian and Paraguayan isolates, whereas OXA-24 producers were almost exclusively found in the USA (Fig. 3).

In contrast, the ST1079 phylogeny comprised only six genomes, three sequenced in this study (Peru) and three from the NCBI database (India, Poland, and one of unknown origin). Despite being reconstructed with the same core-genome approach, ST1079 exhibited markedly greater intercontinental divergence (Table S7). Peruvian isolates differed by just 4–6 core SNPs, whereas non-Peruvian genomes were separated by 2,346–8,328 core SNPs from each other and from the Peruvian clade, consistent with independent introductions from geographically distant regions. All six ST1079 genomes carried OCL3 and KL107, and four (67%) encoded OXA-23. Bootstrap support confirmed the clear separation between the tightly clustered Peruvian subclade and the highly divergent international strains (Fig. 4).

## DISCUSSION

To better understand the genomic context of XDR *A. baumannii* in Peru, we analyzed isolates that were intentionally selected based on their XDR phenotype. All isolates exhibited complete resistance to carbapenems, third-generation cephalosporins, aminoglycosides, and fluoroquinolones, with colistin as the only agent retaining *in vitro* activity. This susceptibility profile is consistent with previous local reports, in which colistin showed high *in vitro* efficacy (~95%) against *A. baumannii*, whereas carbapenems displayed very low susceptibility rates (~2.5%) (13, 19). The persistence of this

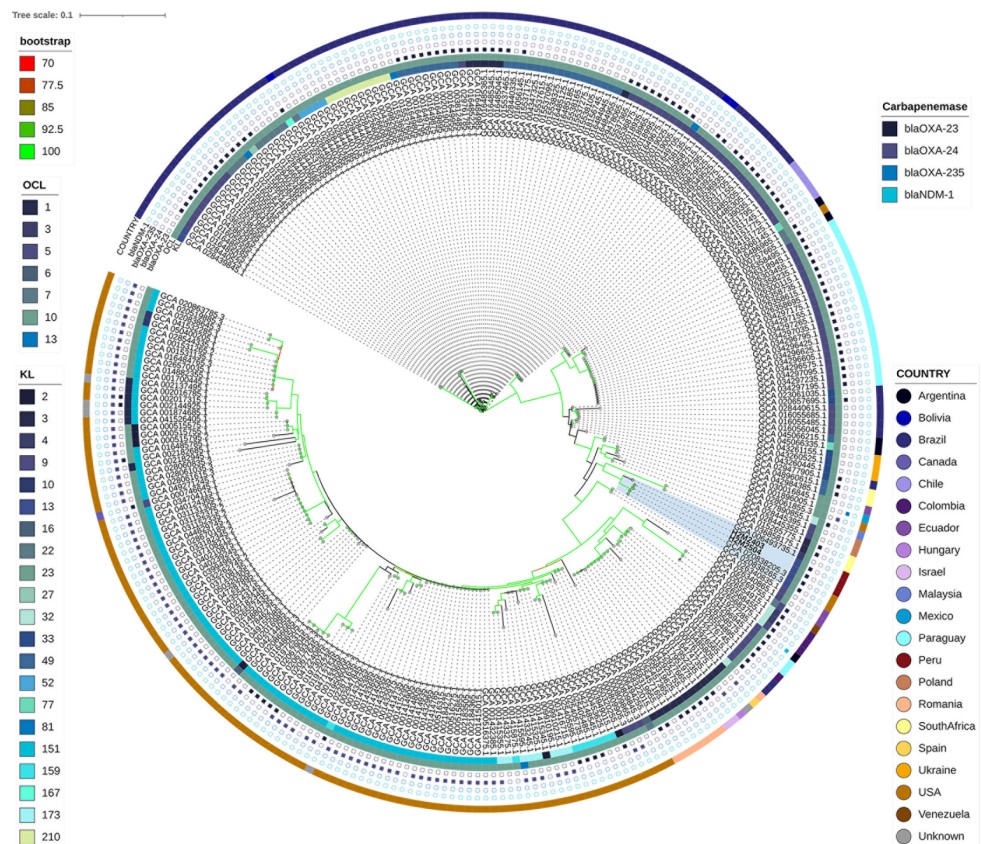

**FIG 3** Global phylogenetic relationships of *A. baumannii* ST79 isolates. The cgSNP-based maximum-likelihood phylogeny depicting worldwide *A. baumannii* ST79 isolates. Bootstrap values (70–100%) are represented along the branches. Metadata tracks indicate KL and OCL locus types, country origin, and the presence of key carbapenemase genes ($bla_{OXA-23}$, $bla_{OXA-24}$, $bla_{OXA-235}$, and $bla_{NDM-1}$). Peruvian genomes are marked to illustrate their phylogenetic position among global ST79 representatives.

resistance pattern underscores the growing concern regarding XDR *A. baumannii* in Peru, with over 97% of isolates reported as carbapenem non-susceptible (13, 19). By comparison, recent large-scale meta-analyses and global surveillance studies report pooled carbapenem resistance rates for *A. baumannii* substantially lower than this figure, with a global meta-analysis reporting resistance rates of 73–76% for carbapenems (20). Other global analyses similarly document wide geographic heterogeneity, with some countries and centers showing extremely high carbapenem-resistant *A. baumannii* burdens while others report much lower rates (11). Taken together, these data indicate that the ~97% carbapenem-resistant *A. baumannii* prevalence observed in Peru is at the upper extreme of the global distribution, highlighting the urgent need for reinforced infection control, antimicrobial stewardship, and genomic surveillance in the country.

Genomic analysis revealed a heterogeneous XDR population dominated by ST2 high-risk clones (73%), with ST79 (11%) and ST1079 (16%) detected less frequently. This distribution aligns with prior Peruvian studies reporting ST2 as the most prevalent lineage in other Lima hospitals (13). Consistent with global epidemiology, the high prevalence of ST2/IC2 among our isolates parallels reports identifying ST2 as the most widely disseminated *A. baumannii* lineage worldwide, in which ST2 and ST1 together accounted for approximately 71% of publicly available genomes (6). However, our findings contrast sharply with those from other South American countries, where ST79, ST25, ST15, or ST1 predominate (7–13).

With respect to carbapenem resistance mechanisms, our ST2 isolates differed from most global ST2 genomes by lacking the $bla_{OXA-23}$ gene (11). Instead, 13/14

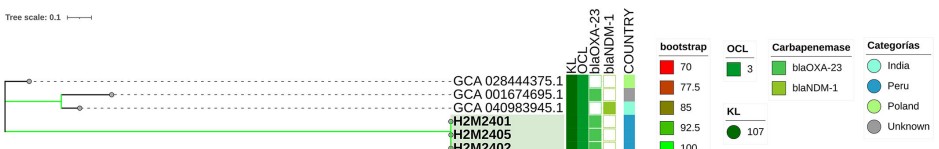

**FIG 4** Global phylogenetic relationships of *A. baumannii* ST1079 isolates. Maximum-likelihood phylogenetic tree based on cgSNPs showing global *A. baumannii* ST1079 isolates. Branch support values (bootstrap, 70–100%) are indicated along the branches. Metadata tracks indicate KL and OCL locus types, country origin, as well as the presence of major acquired carbapenemase genes ($bla_{OXA-23}$, $bla_{OXA-24}$, $bla_{OXA-235}$, and $bla_{NDM-1}$). Isolates sequenced in this study are highlighted to denote their placement within the global ST1079 cluster.

ST2 isolates harbored $bla_{OXA-72}$, a plasmid-borne carbapenemase likely acquired via horizontal gene transfer. Interestingly, despite carrying $bla_{OXA-72}$, some isolates exhibited variable susceptibility to cefepime and ampicillin/sulbactam. Similar findings have been described in $bla_{OXA-72}$-producing *A. baumannii* isolates showing inconsistent β-lactam susceptibility profiles (21, 22). This variability may be attributed to the lower catalytic efficiency of OXA-72 toward extended-spectrum cephalosporins and β-lactam/β-lactamase inhibitor combinations compared with OXA-23, as well as to potential differences in gene expression levels, plasmid copy number, or membrane permeability (22). These observations highlight that the presence of $bla_{OXA-72}$ does not necessarily confer uniform resistance to β-lactams other than carbapenems.

Although *A. baumannii* producing OXA-72 has been reported sporadically in Latin America, they remain far less common than $bla_{OXA-23}$-associated clones, which dominate intra- and intercontinental dissemination (14, 23–26). However, recent data indicate that $bla_{OXA-72}$ is not negligible in the region and has been detected in several South American countries. In Lima, Peru, an analysis of 80 clinical *A. baumannii* isolates collected between 2014 and 2016 revealed that 63.8% (51/80) of carbapenem-resistant ST2 strains and 16.3% (13/80) of ST79 strains carried $bla_{OXA-72}$ (13). In Brazil, although $bla_{OXA-23}$ predominates, $bla_{OXA-72}$ has been reported sporadically, including ST730 isolates in the southern region as well as cases in São Paulo and Recife (27). Similarly, in Ecuador, a clinical outbreak in Guayaquil (2012–2013) revealed that approximately ~90% of the isolates carried $bla_{OXA-72}$ (28). These findings underscore that, although OXA-72 remains less prevalent than OXA-23 across most South American countries, it exhibits a notable geographic distribution and can reach high local frequencies in specific epidemiological settings.

Phylogenomic analyses confirmed the intermixing of isolates from different clinical units within each clade, suggesting multiple independent introductions rather than single-source nosocomial outbreaks. The ST2 lineage comprises two subclusters associated with different KL loci, exhibiting low pairwise SNP distances, which suggests a recent clonal introduction and dissemination from other Peruvian regions (Andean and Amazon). Despite all 14 study genomes being isolated in Lima, this pattern suggests that both ST2 lineages previously identified in other Peruvian regions have recently expanded clonally within the capital. These findings point toward recent interregional introductions into Lima, possibly facilitated by patient transfers or referral networks.

The ST79 lineages formed a local clade with low internal diversity, along with a phylogenetically distant public Peruvian genome. Moreover, our ST79 genotypes matched the South American sublineage, characterized by the presence of $bla_{OXA-23}$ and an XDR phenotype (29). This lineage (IC5) had been reported in several South American countries (Argentina, Brazil, Chile, Colombia, Ecuador, among others), demonstrating wide circulation in hospital, animal, and environmental contexts under a One Health framework (29).

The ST1079 lineages observed exclusively in Peru and absent from public genome databases except for three distantly related isolates (from India and Europe) exhibited low internal SNP variability (7–27 SNPs), supporting the hypothesis of a recent clonal

expansion without clear local introduction. Notably, this lineage harbored the $bla_{OXA-23}$ carbapenemase, raising concerns about a potential outbreak in this clinical setting. The phylogenetic clustering may further support the hypothesis of a single outbreak event.

Taken together, our results supported the emergence of hyperendemic clones from South America in Peru, an ST79 lineage associated with $bla_{OXA-23}$. Furthermore, we observed a high prevalence of ST2 without OXA-23 in Peru, a pattern that aligned with its global expansion but diverged from the typical regional epidemiology. Additionally, we documented the emergence of ST1079 carrying $bla_{OXA-23}$ carbapenemase. These findings revealed a dual epidemiological challenge: (i) high-risk ST2 clones lacking classical carbapenemase OXA-23 and thus evading routine molecular surveillance, and (ii) the introduction and spread of OXA-23-producing ST79 and ST1079 clones. Implementation of WGS-based surveillance in Peru is urgently needed to track the evolution of resistance determinants, detect emerging subclones, and guide both infection control and antimicrobial stewardship strategies in a region where genomic data remain scarce.

This study has several limitations. First, the analysis of a limited number of isolates from a single center may affect the generalizability of our findings; however, they represent the entire XDR *A. baumannii* population identified during the study period in a major Peruvian hospital. Second, phenotypic antimicrobial susceptibility testing was performed using an automated system, which is not a reference method and precluded the testing of newer anti-infective agents typically evaluated for XDR isolates. Third, the genomic analysis relied solely on short-read sequencing. The inclusion of long-read sequencing, even for representative isolates of each clone, would have enhanced the resolution of complex genetic contexts, such as plasmid structures and the precise genomic location of resistance genes. Finally, the absence of dedicated funding limited the scope of the applied methodologies. Despite these limitations, our work provides the first genomic evidence of the establishment of the OXA-23-producing ST79 and ST1079 clones in Peru, offering crucial baseline data for future surveillance and contributing to the understanding of the regional spread of high-risk clones in South America.

## ACKNOWLEDGMENTS

This research was supported by Universidad Nacional Mayor de San Marcos – R. R. N.° 004305-R-2024 and Project number A24011651 – PCONFIGI 2024.

## AUTHOR AFFILIATIONS

[1]Department of Dynamics Sciences, Faculty of Medicine, Universidad Nacional Mayor de San Marcos, Lima, Peru
[2]Department of Clinical Pathology and Pathological Anatomy, Hospital Nacional Dos de Mayo, Lima, Peru

## AUTHOR ORCIDs

Jesus G. M. Pariona  http://orcid.org/0000-0001-9174-0749
Heli Barrón-Pastor  http://orcid.org/0000-0002-4041-4406
José E. Tinedo del Aguila  http://orcid.org/0009-0003-4550-1730
David Santos-Lázaro  http://orcid.org/0000-0003-1450-2039
Doris Huerta-Canales  http://orcid.org/0000-0002-0473-8083
Mario Monteghirfo-Gomero  http://orcid.org/0000-0002-1292-1187
Carolina Cucho-Espinoza  http://orcid.org/0000-0003-3529-4830
Luz Huaroto-Valdivia  http://orcid.org/0000-0002-1197-7250
Yesica Llimpe Mitma de Barrón  http://orcid.org/0000-0001-5420-5093

## FUNDING

| Funder | Grant(s) | Author(s) |
|---|---|---|
| Universidad Nacional Mayor de San Marcos | 004305-R-2024 | Jesus G. M. Pariona |
| | | Heli Barrón-Pastor |
| | | José E. Tinedo del Aguila |
| | | David Santos-Lázaro |
| | | Doris Huerta-Canales |
| | | Mario Monteghirfo-Gomero |
| | | Carolina Cucho-Espinoza |
| | | Luz Huaroto-Valdivia |
| | | Yesica Llimpe Mitma de Barrón |

## AUTHOR CONTRIBUTIONS

Jesus G. M. Pariona, Conceptualization, Data curation, Formal analysis, Investigation, Methodology, Validation, Visualization, Writing – original draft, Writing – review and editing | Heli Barrón-Pastor, Conceptualization, Investigation, Methodology, Project administration, Supervision, Validation | José E. Tinedo del Aguila, Data curation, Formal analysis, Investigation, Resources | David Santos-Lázaro, Data curation, Formal analysis, Investigation, Methodology, Validation | Doris Huerta-Canales, Investigation, Methodology, Resources | Mario Monteghirfo-Gomero, Conceptualization, Investigation, Project administration, Supervision | Carolina Cucho-Espinoza, Investigation, Methodology, Resources | Luz Huaroto-Valdivia, Investigation, Methodology, Resources | Yesica Llimpe Mitma de Barrón, Conceptualization, Formal analysis, Funding acquisition, Investigation, Methodology, Project administration, Supervision, Writing – original draft, Writing – review and editing

## DATA AVAILABILITY

The assembled draft genomes of the 19 XDR Acinetobacter baumannii isolates are publicly available in the NCBI database under BioProject accession number PRJNA1339005 (Genome accessions: JBROGE000000000–JBROGQ000000000, JBRZLA000000000–JBRZLF000000000).

## ETHICS APPROVAL

This study was approved by the Biomedical Research Ethics Committee of Hospital Dos de Mayo (Evaluation No. 051-2024-CEIB-HNDM) and by the Office of Training, Teaching, and Research Support of Hospital Dos de Mayo (Registry No. 17343, Code No. 8009). The study protocol complied with the Code of Ethics of Universidad Nacional Mayor de San Marcos (R.R. No 01992-R-17). Due to the retrospective nature of this study, which used existing bacterial isolates and anonymized clinical data, the ethics committees granted a waiver of informed consent.

## ADDITIONAL FILES

The following material is available online.

### Supplemental Material

**Table S1 (Spectrum03415-25-s0001.xlsx).** NCBI *Acinetobacter baumannii* ST2 genomes and their genome-wide SNP distances to reference genome H2M2302.
**Table S2 (Spectrum03415-25-s0002.xlsx).** Pairwise core-genome SNP distance matrix for 200 global *Acinetobacter baumannii* ST2 genomes, including H2M2302.
**Table S3 (Spectrum03415-25-s0003.docx).** Virulence mechanism groups and associated genes identified in *Acinetobacter baumannii* isolates.

**Table S4 (Spectrum03415-25-s0004.xlsx).** Pairwise core-genome SNP distance matrix for Peruvian *Acinetobacter baumannii* genomes, including 19 isolates from this study and 8 publicly available genomes from NCBI.

**Table S5 (Spectrum03415-25-s0005.xlsx).** Pairwise whole-genome SNP distance matrix for Peruvian *Acinetobacter baumannii* ST2 isolates, including 14 from this study and 4 publicly available genomes.

**Table S6 (Spectrum03415-25-s0006.xlsx).** Core-genome SNP distance matrix of global *Acinetobacter baumannii* ST79 genomes, including two Peruvian study isolates and 278 publicly available genomes from the NCBI database.

**Table S7 (Spectrum03415-25-s0007.xlsx).** Core-genome SNP distance matrix of *Acinetobacter baumannii* ST1079 genomes, including three Peruvian study isolates and three publicly available genomes from the NCBI database.

## Open Peer Review

**PEER REVIEW HISTORY (review-history.pdf).** An accounting of the reviewer comments and feedback.

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
