## [Reviewer comments · Microbiology Spectrum]

Microbiology Spectrum

Whole-genome analysis of extensively drug-resistant *Acinetobacter baumannii* isolates from a Peruvian tertiary hospital reveals the emergence of OXA-23-producing ST79 and ST1079 clones

Jesus Mamani-Pariona, Heli Barrón-Pastor, José Tinedo-Del-Aguila, Elias Santos-Lázaro, Doris Huerta-Canales, Mario Monteghirfo-Gomero, Carolina Cucho-Espinoza, Luz Huaroto-Valdivia, and Yesica Llimpe-Mitma

Corresponding Author(s): Yesica Llimpe-Mitma, Universidad Nacional Mayor de San Marcos Facultad de Medicina de San Fernando

Review Timeline:

Submission Date:	October 27, 2025
Editorial Decision:	November 4, 2025
Revision Received:	November 12, 2025
Accepted:	December 11, 2025

Editor: Cezar Khursigara

Reviewer(s): The reviewers have opted to remain anonymous.

Transaction Report:

DOI: <https://doi.org/10.1128/spectrum.03415-25>

Re: Spectrum03415-25 (Whole-genome analysis of extensively drug-resistant *Acinetobacter baumannii* isolates from a Peruvian tertiary hospital reveals the emergence of OXA-23-producing ST79 and ST1079 clones)

Dear Dr. Yesica Llimpe-Mitma:

Thank you for the privilege of reviewing your work. Below you will find my comments, instructions from the Spectrum editorial office, and the reviewer comments.

I am pleased to inform you that your manuscript has been editorially accepted for publication. However, there are a few additional questions in the submission form that need to be answered before the final decision. Once these are completed, please return your submission so that I can move your paper forward to acceptance.

Sincerely,
Cezar Khursigara
Editor
Microbiology Spectrum

Re: Spectrum03415-25R1 (Whole-genome analysis of extensively drug-resistant *Acinetobacter baumannii* isolates from a Peruvian tertiary hospital reveals the emergence of OXA-23-producing ST79 and ST1079 clones)

Dear Dr. Yesica Llimpe-Mitma:

Your manuscript has been accepted, and I am forwarding it to the ASM production staff for publication. Your paper will first be checked to make sure all elements meet the technical requirements. ASM staff will contact you if anything needs to be revised before copyediting and production can begin. Otherwise, you will be notified when your proofs are ready to be viewed.

Sincerely,
Cezar Khursigara
Editor
Microbiology Spectrum